# Chitosan Extracted from the Biomass of *Tenebrio molitor* Larvae as a Sustainable Packaging Film

**DOI:** 10.3390/ma17153670

**Published:** 2024-07-25

**Authors:** Chacha Saidi Mwita, Riaz Muhammad, Ezekiel Edward Nettey-Oppong, Doljinsuren Enkhbayar, Ahmed Ali, Jiwon Ahn, Seong-Wan Kim, Young-Seek Seok, Seung Ho Choi

**Affiliations:** 1Department of Biomedical Engineering, Yonsei University, Wonju 26493, Republic of Korea; chachasaidimwita@yonsei.ac.kr (C.S.M.); riaz@yonsei.ac.kr (R.M.); ezekieledward@yonsei.ac.kr (E.E.N.-O.); doji0704@yonsei.ac.kr (D.E.); jwyd010601@gmail.com (J.A.); 2Department of Electrical Engineering, Sukkur IBA University, Sukkur 65200, Pakistan; 3Department of Agricultural Biology, National Institute of Agricultural Sciences, Rural Development Administration, Wanju 55365, Republic of Korea; tarupa@korea.kr; 4Gangwon-do Agricultural Product Registered Seed Station, Chuncheon 24410, Republic of Korea; 5Department of Integrative Medicine, Major in Digital Healthcare, Yonsei University College of Medicine, Seoul 06229, Republic of Korea

**Keywords:** food packaging, chitosan films, biomass, cross-linking, plasticizer, mealworm shells

## Abstract

Waste from non-degradable packaging materials poses a serious environmental risk and has led to interest in developing sustainable bio-based packaging materials. Sustainable packaging materials have been made from diverse naturally derived materials such as bamboo, sugarcane, and corn starch. In this study, we made a sustainable packaging film using chitosan extracted from the biomass of yellow mealworm (*Tenebrio molitor*) shell waste. The extracted chitosan was used to create films, cross-linked with citric acid (CA) and with the addition of glycerol to impart flexibility, using the solvent casting method. The successful cross-linking was evaluated using Fourier-Transform Infrared (FTIR) analysis. The CA cross-linked mealworm chitosan (CAMC) films exhibited improved water resistance with moisture content reduced from 19.9 to 14.5%. Improved barrier properties were also noted, with a 28.7% and 10.2% decrease in vapor permeability and vapor transmission rate, respectively. Bananas were selected for food preservation, and significant changes were observed over a duration of 10 days. Compared to the control sample, bananas packaged in CAMC pouches exhibited a lesser loss in weight because of excellent barrier properties against water vapor. Moreover, the quality and texture of bananas packaged in CAMC pouch remained intact over the duration of the experiment. This indicates that adding citric acid and glycerol to the chitosan structure holds promise for effective food wrapping and contributes to the enhancement of banana shelf life. Through this study, we concluded that chitosan film derived from mealworm biomass has potential as a valuable resource for sustainable packaging solutions, promoting the adoption of environmentally friendly practices in the food industry.

## 1. Introduction

Food preservation is a significant challenge. Globally, there is a need to address this issue, with a target to reduce food wastage per capita by 50% in both commercial and consumer contexts by 2030 [1,2]. Food packaging is crucial for preserving food against biological, chemical, and physical threats. With a growing emphasis on maintaining food quality, there is increasing scrutiny over the selection of packaging materials [3,4]. 

Currently, approximately 350–380 million tons of plastic are generated each year, with packaging accounting for the largest portion of plastic waste at 61% [5]. The environment is negatively affected by the heavy use of petrochemical materials for food packaging. Such materials are neither environmentally sustainable, biodegradable, nor recyclable. To achieve sustainability and cleaner production objectives, it is necessary to reduce packaging waste. As a result, there is increasing interest in creating new sustainable bio-based packaging materials derived from biological sources as an alternative to synthetic or non-biodegradable plastics to lessen their environmental impact. 

In recent years, active packaging has garnered significant interest among researchers and professional experts in the field of food packaging [4]. This involves integrating active ingredients into the packaging material. Such integration can change how food is stored, affect its metabolic processes, and enhance its protective properties, ultimately leading to improved quality and an extended shelf life [6]. Traditionally, petrochemical polymers, notably plastics, have been the preferred materials for active packaging applications in industries such as food, cosmetics, and pharmaceuticals due to their affordability and excellent mechanical properties [7]. However, these polymers are non-biodegradable, contributing to environmental concerns. 

As an alternative approach, natural biopolymers, specifically proteins and polysaccharides, have garnered increasing attention for creating biodegradable, and biocompatible food packaging materials [8]. Commonly, these materials include starch [8], pullulan [8], pectin [9], cellulose [10], carrageenan [11], alginate [12], and chitosan [13]. Chitosan, in particular, holds promise as a food packaging due to its structural versatility, biodegradability, antibacterial capability, biocompatibility, antimicrobial, and antioxidant properties [14]. Moreover, chitosan is recognized as a safe material for food preservation because of its nontoxic, non-carcinogenic, and non-immunogenic properties. It is designated as Generally Recognized as Safe (GRAS) by FDA (Food and Drug Administration) in 2013, indicating that it has been acknowledged by qualified experts as a safe food additive [15]. Furthermore, because of the remarkable film-making capabilities of chitosan, it has been studied for diverse applications in the fields of biomedical, chemical, and food domain [16].

In recent studies, chitosan has been shown to be a promising material for developing biodegradable packaging for food. It can help prolong food shelf life, minimize reliance on chemical preservatives, and decrease the production of plastic waste generated from petrochemical biopolymers. It should be noted that the invasion of spoilage and food poisoning bacteria is the main cause related to food deterioration among intrinsic and extrinsic factors. Therefore, the delay and prevention of bacterial growth on food products are crucial for preparing active packing material. Chitosan has intrinsic antimicrobial properties due to active amino groups [17]. 

Chitosan is produced through the deacetylation of chitin, the second most commonly available polysaccharide found in nature and known since 1884 [18]. Chitin is a polymer made of poly(β-(1→4)-N-acetyl-d-glucosamine)). It is derived from natural sources such as shrimp, crab, lobster, krill shells, fungi, arthropods, and mealworms [19]. This makes chitosan, derived from chitin, a widely accessible polysaccharide. Chitosan, in its solid phase, exhibits semi-crystalline properties and readily dissolves in weak organic acids for instance tartaric, citric, acetic, formic, malic, or lactic acid [20,21]. Chitosan films have been successfully tested in experiments on a variety of dairy fresh and processed products, demonstrating their ability to prolong their shelf life and improve their quality by shielding them from contamination and microbial deterioration. Moreover, chitosan films exhibit moderate values of oxygen and water permeability [22]. Some of the benefits of using chitosan films include high resistance to fracture, high toughness levels, strong property values, long service life cycles, high structural design flexibility, especially in packaging, and reduced food respiration rates [22]. To regulate their mechanical and barrier properties, chitosan films can incorporate plasticizers and cross-linkers [23]. Adding cross-linkers enhances the barrier and mechanical characteristics of chitosan structures, while plasticizers decrease intermolecular interactions in the chitosan network, thereby improving flexibility and mobility between the network segments [24,25].

Recently, many food packaging studies have reported chitosan and modified chitosan as promising active food packaging materials. Srasti Yadav et al. [26] studied chitosan films modified with ZnO nanoparticles and gallic acid for potential use as a packaging material for food. The authors reported that adding ZnO-gallic acid greatly improved the UV light-blocking characteristics, mechanical strength, moisture, and oxygen resistance, as well as the film’s antioxidant and antimicrobial activities. The most effective composition included 70 mg of ZnO-gal, demonstrating the best overall performance. The researchers further demonstrated that the developed film increased the storage duration of fresh food products such as black grapes, apples, mangoes, and tomatoes, providing an environmentally friendly alternative to conventional packaging. M. El Mouzahim et al. [27] performed research for assessing the influence of incorporating *Ficus carica*-mediated AgNPs (silver nanoparticles) and kaolin clay into chitosan-based films for packaging food to preserve the freshness of apple slices. They observed that chitosan/silver nanoparticle film had better antibacterial and mechanical properties compared to chitosan/kaolin clay film. However, the latter showed enhanced vapor resistance characteristics and was found to be biodegradable. The chitosan/kaolin film successfully maintained the freshness of apple slices, without discoloration, retaining moisture, and preserving important bioactive compounds like antioxidants and polyphenols. Likewise, Sofia M. Costa et al. [28] assessed the possibility of CNC (chitosan/cellulose nanocrystal) films as active packaging to extend the freshness of meat for a longer period. The solvent casting technique was employed to form the film. CNC integration increased the oxygen barrier and thermal properties, whereas no significant difference was observed in water vapor permeability performance. Meat wrapped in CNC film showed the lowest TVB-N levels following a two-week storage period. Shengye Dong et al. [29] reported the fabrication of a multifunctional film with antibacterial and pH-responsive properties by integrating anthocyanin-rich extract from potatoes with purple pigment chitosan nanoparticles loaded with quercetin into a polymer matrix consisting of agar and sodium alginate. This film was designed to monitor and prevent shrimp deterioration. Tingting Zheng et al. developed pH-sensitive films composed of chitosan, collagen, mulberry extract-containing (ZnO/ME/CC), and ZnO nanoparticles for the purpose of monitoring the freshness of pork [30]. 

This study explores the influence of incorporating glycerol for plasticization and citric acid for cross-linking on the performance of chitosan-based films fabricated for use as food packaging material to enhance the shelf life of bananas. The as-prepared film underwent characterization through Fourier transform infrared spectroscopy (FTIR). Their properties were compared to those of unmodified chitosan films. Moreover, the chemical and physical properties of the chitosan-based films were evaluated, including solubility and absorption of water, content of moisture in the films, and barrier to inhibit the passage of water vapor. This marks the initial investigation of CA cross-linked mealworm chitosan (CAMC) films for use in packaging fresh goods, including their influence on the quality of stored bananas, to prolong the storage time of bananas.

The specific aims of this study are to introduce new means of food packaging and to demonstrate that citric acid-incorporated chitosan films are a sustainable alternative for food packaging in terms of improved water resistance, enhanced barrier properties, and extended shelf life of bananas.

## 2. Materials and Methods

### 2.1. Materials

Shells of commercial yellow mealworm (*Tenebrio molitor*) were obtained from the smart factory located at Gangwon Provincial Agricultural Products Center in Chuncheon, Republic of Korea. Since 2020, the yellow mealworm beetle (*Tenebrio molitor*) has been continuously bred and managed for mass production at this Smart Factory. This facility utilizes advanced technologies, including IoT-powered sensors and artificial intelligence, to optimize the breeding and feeding processes in real time. By continuously monitoring and adjusting the feeding process, the factory ensures optimal growth, and selective feeding, and improves the overall health of the mealworms. The breeding method is based on strict genetic evaluation and species selection criteria, with a focus on high fertility, rapid growth, and high survival rates. To enable successful outbreeding with different strains, they employ big data analysis to select pupae based on weight and egg output. This method significantly enhances egg production, larval survival rates, and individual larval weight. This smart farming approach facilitates the mass production of yellow mealworm (*Tenebrio molitor*), achieving an annual yield of 10 tons per year. This high yield results in substantial biomass waste production, specifically mealworm shells, which are used as the primary material for chitosan extraction in this study.

Sodium hydroxide 98.0% was purchased from OCI-Company Ltd, Seoul, Republic of Korea. Acetic acid (Glacial 99.5%) was acquired from DAEJUNG Reagents Chemicals & Metals Co., Ltd., Siheung, Republic of Korea. Glycerol assay ≥ 99.5% purchased from SIGMA-ALDRICH, Darmstadt, Germany. Miracloth 475855-1R filter paper bought from EMD Millipore Corp 290 Concord Rd, Billerica, MA, USA. All substances and chemicals used in the experiment were of analytical quality. The water resistance characteristics of the fabricated chitosan-based films were investigated using a 3D-printed component designed with computer-aided design (CAD) software (Autodesk Fusion 360, Autodesk, Inc., San Francisco, CA, USA) and printed with a print precision of ±0.2 mm. The 3D printing was performed utilizing a 3D printer (model Flashforge Guider II, Zhejing Flashforge 3D Technology Co., Ltd., Jinhua, China). Polylactic Acid (PLA) material, featuring a filament diameter of 1.75 mm, was employed for the printing process.

### 2.2. Pretreatment of Mealworm Shells Biomass

Initially, the yellow mealworm shells (*Tenebrio molitor*) waste was manually cleaned to remove mealworm remnants or other impurities. To remove dirt particles and unwanted materials, the collected mealworm shells were carefully washed three times using hot water. Once cleaned, the shells were dried in an oven. Subsequently, the dried samples were finely processed using a grinding machine and transferred to a sealed container for storage.

### 2.3. Extraction of Chitosan from Mealworm Shell Biomass

The chitosan extraction was carried out using a chemical extraction method, involving three primary processes: demineralization, deproteination, and deacetylation, to obtain chitosan.

#### 2.3.1. Demineralization

A 5 g sample of mealworm shells was subjected to demineralization using 1N acetic acid with a shell concentration of 3 wt%. The demineralization process was conducted for a duration of 12 h at a temperature of 60 °C, while continuously stirring at 400 rpm. To reduce foaming caused by gas generation due to the presence of calcium carbonate in the shells, acid was gradually added. Carbon dioxide is produced when the acid reacts with calcium carbonate. After demineralization, the sample was thoroughly rinsed with deionized (DI) water. The material was then dried for 48 h at a temperature of 60 °C.

#### 2.3.2. Deproteination

The demineralized mealworm shell powder was deproteinized using 2% NaOH. The process lasted 12 h at 60 °C, with continuous stirring at 400 rpm, followed by thorough washing and drying. The outcome of this process yielded chitin.

#### 2.3.3. Deacetylation

To carry out deacetylation, the chitin underwent treatment with a concentrated alkali, specifically 50% sodium hydroxide. The deacetylation process occurred over a period of 5 days at a temperature of 120 °C, with a 3 wt% chitin concentration and continuous stirring at 400 rpm. The resulting chitosan was then passed through miracloth and meticulously cleaned using DI (deionized) water to ensure purity.

### 2.4. Fabrication of Film

The solvent casting method was employed to fabricate chitosan-based films. A chitosan solution was prepared by adding 8 g of as-prepared mealworm-extracted chitosan to 200 mL of a 1% *v*/*v* acetic acid aqueous solution, resulting in a final chitosan concentration of 4% *w*/*v*. To achieve thorough dissolution, this mixture was agitated continuously overnight at 60 °C using a magnetic stirrer with a 3 cm stirring bar at 300 rpm. The solution obtained was used to fabricate the plain mealworm chitosan (MC) film. The solution was degassed using a desiccator with a vacuum pump maintaining a pressure of 740 mmHg to eliminate any trapped air bubbles. The prepared chitosan solution, totaling 200 mL, was subsequently poured onto a 32 by 23 cm Petri dish and left for drying for 48 h at 60 °C in a heating oven. The temperature was monitored using an internal probe. Before conducting the characterizations, the films were stored in an enclosed desiccator at 25–30% relative humidity and 25 °C to avoid any potential absorption of moisture. 

Similar procedures were applied in the fabrication of glycerol-plasticized and CA-cross-linked mealworm chitosan (CAMC) films. A chitosan solution was prepared by adding 8 g of as-prepared mealworm-extracted chitosan to 200 mL of a 1% *v*/*v* acetic acid aqueous solution with a final chitosan concentration of 4% *w*/*v*. The solution was stirred vigorously at 220 rpm for 12 h. After allowing the chitosan to dissolve overnight in the acetic acid solution, a mixture of citric acid in a 1:0.5 ratio to chitosan was prepared and left to blend for 6 h. Following the addition of citric acid, glycerol was introduced at a ratio of 0.1:1 relative to chitosan weight. After stirring for an additional 3 h, the glycerol and citric acid mixture with chitosan solution was poured into 32 by 23 cm dish containers. The containers were left to dry at 90 °C in a ventilated oven for 24 h to initiate the cross-linking process, with the temperature monitored using an internal probe. Finally, the films were stored in a sealed desiccator at 25–30% relative humidity and 25 °C to prevent potential moisture absorption.

### 2.5. Measurement of Moisture Content

The moisture content of a dry film, signifying the quantity of moisture it absorbs until a balance is reached between its water content and that of the ambient environment, is referred to as the film’s moisture content [31]. To determine it, the films were sectioned into 40 × 40 mm squares and weighed. W_1_ denotes the initial weight of the moisture-containing films. These films were heated in laboratory grade oven at 120 °C till they reached a constant weight, with W_2_ representing the weight of the dried film. We use the below equation to determine the moisture content: (1)$$\mathrm{Water}\;\mathrm{Content}\;(\%)=\frac{{W_{1}-W}_{2}}{W_{1}}$$

### 2.6. Measurement of Water Absorption

The quantification of moisture uptake in a film can be determined by the volume of moisture it absorbs until it reaches a state of equilibrium [32]. This measurement is expressed as a percentage relative to the weight of the completely dried film. The film, fully dried and represented by weight W_d_ from the preceding step, was immersed in 25 mL of distilled water to evaluate water absorption. The weight increase was monitored until equilibrium was reached. W_w_ symbolizes the ultimate weight of the film when it reaches saturation with water at equilibrium. The calculation of moisture uptake was carried out using the following formula:(2)$$\mathrm{Water}\;\mathrm{Absorption}\;(\%)=\frac{{W_{w}-W}_{d}}{W_{d}}$$

### 2.7. Assessment of Water Solubility

The film’s water solubility is evaluated by gauging how much of the film’s solid content dissolves when exposed to water [33]. We evaluated the dissolvability of the films using modified versions of established methods. In summary, 40 × 40 mm square sections of the films were arranged in a vacuum desiccator to ensure the thorough elimination of any residual moisture content. These films were consistently weighed until they reached a point where the weight indicated complete drying, termed as the beginning dry mass W_id_. Next, the fabricated films were placed in a glass container with 40 mL of DI water and stirred continuously for 24 h at 30 °C. Afterward, the films were taken out of the beakers and put to dry at 110 °C until they cooled down and reached their final dry weight W_fd_. The water solubility of produced chitosan-based films was calculated using Equation (3).
(3)$$\mathrm{Water}\;\mathrm{Solubility}\;(\%)=\frac{{W_{\mathrm{id}}-W}_{\mathrm{fd}}}{W_{\mathrm{id}}}$$

### 2.8. Determination of Water Vapor Permeability

The permeability of MC and CAMS films to water vapor was evaluated utilizing the wet cup procedure described by ASTM E96-95 [7]. For analyzing the permeability of the fabricated chitosan films to water vapor, we utilized a wide-mouth cup with a 50 mm diameter and added 50 mL of distilled water. It is worth noting that we maintained a headspace of less than 5 mm between the water’s surface and the film. 

To minimize water evaporation through the cup’s edges, we sealed it with a waterproof lid and regulated the temperature at 30 °C for control. The closed system was conditioned for 3 h to make sure that the whole system would reach the required temperature before the experiment began. During this time, the permeability chamber was prepared containing silica, and the cup was filled with water. When the temperature of the system reached 30 °C, the top lid of the waterproof cup was opened, and the first sample was inserted between two gaskets by screwing the lid down to have it sealed tight. The sample was positioned accurately and carefully without tearing or folding it while covering the entire opening of the cup. The cup was then put back in the center of the weight balance. All samples were tested at a temperature of 30 °C for at least 3 h, allowing sufficient time for the water vapor flow to reach the equilibrium point. The weight of the cup was measured with an accuracy of 0.0001 g every 30 min. Equations (4)–(6) determine the water vapor transmission rate (WVTR), permeance, and permeability (WVP) of the fabricated MC and CAMC films, respectively.
(4)$$\mathrm{WVTR}\left(\frac{g}{s\cdot m^{2}}\right)=\frac{\Delta w}{\Delta t\cdot A}$$
(5)$$\mathrm{Permeance}\;\left(\frac{g}{s\cdot m^{2}\cdot \mathrm{Pa}}\right)=\frac{\mathrm{WVTR}}{\Delta P}$$
(6)$$\mathrm{Permeance}\cdot \mathrm{thickness}=\mathrm{WVP}\left(\frac{g}{s\cdot m\cdot \mathrm{Pa}}\right)\;\mathrm{WVP}\left(\frac{g}{s\cdot m\cdot \mathrm{Pa}}\right)=\mathrm{Permeance}\cdot \mathrm{thickness}$$
where the flux (Δw/Δt) represents the rate at which the cell loses weight per unit time determined by the slope of the cup’s weight loss versus time; A denotes the total exposed area, calculated using the diameter of the mouth of the cup, while ΔP represents the difference in water vapor pressure at 30 °C. The experiment was conducted three times under the same conditions where the headspace was fully saturated with water vapor and the environment was completely dried using silica gel. 

### 2.9. Measurement of Film Thickness

A digital caliper (model DC150-1, CAS Co., Ltd., Yangju, Republic of Korea) was used to measure the thickness of the film, and five arbitrary measurements were made for each sample. The average of these measurements was then calculated as the mathematical mean.

### 2.10. Measurement of Water Contact Angles

To determine contact angles, 3 μL of distilled water was dispensed onto 2 × 2 cm^2^ film samples using a micropipette. The samples were then firmly attached to a glass slide with a glue stick. The slide was mounted on a stage, and a camera was positioned perpendicular to the surface to capture an image of the water drop. The captured image was analyzed using the Contact Angle plugin of the open-source ImageJ software (Version 1.54h, NIH, Bethesda, MD, USA). The contact angle measurements were reported as the average of three independent readings.

### 2.11. FTIR Analysis

Fourier transform infrared spectroscopy (FTIR) was employed to analyze the chemical composition of the films and investigate potential interactions with citric acid. FTIR analysis of MC and CAMC films was conducted using a Bruker Alpha Optics FTIR (model Alpha II, Bruker, Billerica, MA, USA) in transmission mode. The samples were cut into dimensions of 1 cm by 1 cm. For measurement, each sample was placed under the FTIR probe with the smooth side facing down, and the probe was mechanically fixed onto the film surface. The measurements were repeated five times for each sample, capturing spectral information over a range of wavenumbers from 400 to 4000 cm^−1^ at a resolution of 4 cm^−1^, with an accumulation of 64 scans per sample. The OPUS Spectroscopy software (Version 8.8, Bruker, Billerica, MA, USA) was used to record and analyze the Fourier spectra of the samples. All spectra were recorded at a room temperature of 25 °C.

### 2.12. Evaluation of CAMC Films for Banana Packaging

The wrapping performance of fabricated films was examined for bananas. The films were fashioned into pliable pouches measuring 110 mm in length and 60 mm in width. Three sets of fresh bananas were employed for the study: unpackaged, packaged in polyethylene bags, and enclosed in CAMC films. All banana samples were stored under identical conditions to ensure a fair comparison: they were exposed to air at an ambient temperature of 25 °C with a relative humidity (RH) of approximately 60–65%. The RH was controlled using silica gel and monitored using a humidity meter, ensuring consistent humidity levels throughout the storage period. This control condition is critical as it represents typical room temperature storage, providing a consistent baseline against which the effectiveness of the CAMC films can be measured. This allows us to investigate the effect of CAMC films on banana preservation more effectively. A similar methodology has been used in several studies to investigate how packaging affects the storage time of various food products, for instance, fresh produce, bread slices, and meat [34,35]. For the first time, we have investigated the real-world applications of glycerol-plasticized and citric acid-cross-linked mealworm-extracted chitosan films for banana preservation. 

### 2.13. Determination of Weight Loss (%) of Banana

To assess weight loss, banana samples were initially marked and weighed using a digital balance (model ADB 200-4, KERN & SOHN GmbH, Balingen, Germany). These banana samples were weighed again at the beginning of the experiment and subsequently at 1-day intervals over a period of 10 days. The percentage of weight loss for the fruits was computed using the formula in Equation (7), wherein W_1_ and W_2_ represent the initial weights and weights at specific intervals, respectively.
(7)$$\mathrm{Weight}\;\mathrm{Loss}\;\left(\%\right)=\frac{{W_{1}-W}_{2}}{W_{1}}\;\times \;100$$

### 2.14. Surface Characterization Using Scanning Electron Microscopy (SEM)

The surface morphology of the fabricated MC and CAMC films was characterized using a Schottky Field Emission Scanning Electron Microscope (model JSM-7800F, Japan Electron Optics Laboratory Ltd., Tokyo, Japan). A sample of 1 cm by 1 cm was cut from the film and sputter-coated with a 10 nm thick layer of platinum.

### 2.15. Statistical Analysis

Data are expressed as the mean ± standard deviation (SD; n ≥ 3). The collected data from three experiments on banana weight loss were analyzed statistically using OriginPro 2016 (OriginLab, Northampton, MA, USA). A one-way analysis of variance (ANOVA) was used to measure the significance of using different packaging materials (i.e., CAMC, polyethylene, and unpackaged). Tukey’s test was applied to determine the significant differences among the data sets. The significance of the difference between pairs of means was compared at a 5% level of probability (*p* ≤ 0.05).

## 3. Results and Discussion

### 3.1. Fabrication of MC and CAMC Films

Figure 1a shows the schematic representation of the procedure for chitosan extraction from mealworm shells. We began the procedure with biomass derived from mealworm shell waste, which served as the raw material. This raw material underwent a pretreatment procedure, wherein the obtained flakes, known as mealworm shells, are finely pulverized into powder form. Pulverization increases the surface area, which improves the efficiency of subsequent extraction steps. The powder was washed multiple times to remove all impurities and then dried thoroughly. Thorough washing ensures the removal of any contaminants that could affect the extraction process. 

Subsequently, we conducted purification procedures to obtain chitin. The first step in this process was demineralization, which removed the minerals present in the mealworm shell waste. Minerals affect the quality and purity of the extracted chitin, which is why demineralization is so important. We achieved this by dissolving the powder in an acetic acid solution. Acetic acid is utilized because it dissolves calcium carbonate and other minerals efficiently. 

Following demineralization, we carried out deproteinization through an alkali digestion process using sodium hydroxide solution, which effectively removes the inherent proteins in the material. The present protein molecules are linked to the chitin structure via glycosidic bonds. Purified chitin requires deproteinization since proteins are firmly attached to the chitin structure. Sodium hydroxide breaks these bonds, allowing for their removal. To obtain chitosan from chitin, we performed further treatment using a higher concentration of sodium hydroxide solution in a procedure termed deacetylation. The deacetylation procedure converts the N-acetyl groups present in the chitin structure into amino groups, which is how chitosan is formed. 

Figure 1b shows the chemical structures of chitin and chitosan and the conversion of chitin into chitosan by deacetylation. Chitosan is a well-known glycoprotein whose structure is formed by randomly situated N-acetyl-d-glucosamine and d-glucosamine groups linked by β-1,4-glucosidic linkages [31]. Chitosan is made by partly deacetylating chitin [32]. Chitosan exhibits insolubility in both water and solvents made from organic matter; however, it readily dissolves in acidic solutions with a pH less than 6.5. This solubility is due to the protonation of the glucosamine amino unit (-NH_2_), which converts into glucosamine-NH_3_^+^, a soluble cationic form [33]. Figure 1c shows digital photos of mealworms, mealworm shells, chitin during the extraction process, and the extracted chitosan. The rightmost part of Figure 1c shows the mealworm-extracted chitosan film. The film was prepared using the solvent casting method.

Figure 2 shows the process of forming cross-linked chitosan film using the solvent-casting method. The process of making the chitosan solution involves dissolving the chitosan in an acetic acid solution, as seen in the upper part of Figure 2. Acetic acid is commonly used to completely dissolve chitosan by breaking down its crystalline structure. After preparing the chitosan solution, citric acid is added. Citric acid acts as a cross-linking agent, which enhances the polymer matrix of the chitosan solution, improving its structural integrity. Additionally, glycerol is added to the solution to improve the flexibility of the film. Glycerol functions as a plasticizer, reducing intermolecular forces within the polymer matrix and increasing its flexibility and elasticity. After adding citric acid and glycerol and stirring for three hours, the solution is cast onto a Petri dish. Subsequently, the Petri dish is kept in an oven overnight to allow the solution to dry. Finally, after drying, the citric acid cross-linked mealworm chitosan (CAMC) film is obtained.

Similarly, we used the same procedure for preparing the pure mealworm chitosan (MC) film, except for the steps of adding citric acid and glycerol to the solution. After preparing the chitosan solution, it was spread on a Petri dish and put in an oven to fabricate film. 

### 3.2. Characterization of MC and CAMC Films

Figure 3a shows the produced CA cross-linked mealworm chitosan film. To create a pouch for packaging bananas, the CAMC film was first cut into rectangular sheets of appropriate size. The film sheets were then folded into the shape of a pouch. As shown in Figure 3b, the pouch’s open ends were carefully sealed with scotch tape to prevent moisture or air particles from entering or leaving the package. Through this process, the CAMC film’s structural stability and flexibility are also highlighted, demonstrating its applicability to food packaging applications. This flexibility is due to the plasticization incorporated by glycerol and the stability provided by the cross-linking of citric acid.

As depicted in Figure 4, FTIR analysis was performed to confirm the cross-linking of CA. All distinct bands of the chitosan structure are evident in the FTIR spectrum of MC films. The large absorption band between 3200 and 3300 cm^−1^ corresponds to vibrations involving stretching of -NH and -OH. The amide vibrations of C=O stretching are associated with the absorption band at 1533 cm^−1^. The peak at 1420 cm^−1^ is attributed to the bending vibration of C-H bonds, while the peak at 2950 cm^−1^ represents the stretching vibrations of C-H bonds. Meanwhile, the absorption band at 1120 cm^−1^ is attributed to C-O stretching within the ring structure of chitosan.

At room temperature, CAMC exhibited a nearly identical absorption band pattern, with the exception of the stretching band associated with the ester C=O group observed at 1730 cm^−1^ [36,37]. The ester bond formed via cross-linking is shown by a new peak found in the FTIR spectra of the CAMC film, which is located at 1730 cm^−1^. This finding validates that there is no untreated citric acid available in the film and the new band is due to the cross-linking. Figure 5 shows the proposed cross-linking scheme. In the presence of CA, chitosan loses a hydrogen ion, resulting in the protonated form of chitosan. The protonated chitosan undergoes reactions with citric acid to make cross-linked structures via intramolecular and intermolecular bond creation.

### 3.3. Water Resistance Characteristics of MC and CAMC Films

The water resistance characteristics of the fabricated MC and CAMC films are crucial factors for their use in food packaging applications. This was evaluated by determining their moisture content, solubility, and absorption rates. The capacity of MC and CAMC films to dissolve is determined by factors such as the behavior of the film’s surface, wettability, and their composition. Table 1 presents water solubility, water absorption, and moisture content values for both MC and CAMC films.

In this study, the water solubility, water absorption, and moisture content for mealworm chitosan (MC) film were 37.55 ± 1.54%, 59.74 ± 3.26, and 19.99 ± 0.45%, respectively, which is comparable to previous reports on chitosan film [38]. Moreover, we found out that the chitosan films’ moisture content and water absorption decrease because of cross-linking with CA. For CAMC film, the moisture content and water absorption values dropped by roughly 27% and 39%, respectively. Cross-linking likely creates a more compact and stable polymer network structure within the chitosan matrix, which limits the number of hydrophilic groups that can interact with water molecules. During cross-linking with CA, covalent bonding is formed by these hydrophilic groups thus making them unavailable for any interaction with water anymore. The interactions between water and polysaccharides led to a reduction in water absorption and moisture content measurements for the CAMC film.

### 3.4. Moisture Barrier Characteristics of MC and CAMC Films

The WVTR (water vapor transmission rate) and WVP (water vapor permeability) of the fabricated MC and CAMC films have also been investigated. WVP and WVTR are crucial moisture barrier properties that determine the effectiveness of MC and CAMC films as functional materials for food packaging purposes. The barrier and mechanical characteristics of MC and CAMC films directly influence the value of WVTR. Moisture permeation into packaging materials considerably influences the preservation environment of food and improves the storage time of fresh products. Factors like free volume, crystallinity, and molecular interactions mainly determine the permeability of the film. The role of the moisture-resistant layer in packaging fresh products is crucial, as it acts as a barrier preventing moisture from escaping and reducing the hydration rate of the food. WVP measures the rate of vapor transmission through the film in terms of grams per second per square meter Pascal’s (g/s·m·Pa). Following a gravimetric approach, the most employed technique for investigating WVP is the standard ASTM E96 method. We created a water cup with the dimensions explained in Section 2.8. using 3D printing (see Figure 6a). To measure water vapor permeability, we placed the chitosan film between the gaskets and closed the top lid, as shown in Figure 6b.

Since there was distilled water inside the cup, its vapor pressure was affected by the specific temperature of the experiment, which was set to 30 °C. At this temperature, the vapor pressure generated inside the chamber creates a gradient that forces water vapor to pass through the film, as shown in Figure 6c. At regular intervals, we measure the weight of the cup to quantify the weight loss. Then, we calculate the WVP and WVTR using the measured weight loss, temperature, and dimensions of the test cup. 

Table 2 shows the WVP and WVTR data for MC films against CAMC films. As shown in the table, WVTR and WVP exhibit a reduction of 10.2% and 28.7%, respectively. Reddy and Yang’s investigation shows comparable outcomes for starch films that are incorporated with glycerol for plasticization and cross-linked with CA [39]. 

The intramolecular distance of the chitosan matrix was widened by the plasticizer glycerol, which produced films with decreased density [40]. The cross-linking agent may cause the chitosan film’s levels of hydrophilic free amine functional groups to drop. Furthermore, a compact chitosan matrix limits the swelling of the film, hence decreasing the quantity of vapor molecules that can pass from the chitosan-based film [39]. Therefore, decreasing the process of water vapor transport.

The contact angles of the produced MC and CAMC films were tested to determine their hydrophobic properties. A water droplet is shown in Figure 7 on the surface of the fabricated MC and CAMC film, with a green line around it representing the measured contact angle. The surface characteristics of the chitosan film were evaluated to assess their suitability for food packaging, focusing on their ability to provide information about surface properties. This evaluation involved measuring contact angles, which were found to be 75.9° for MC and 89.3° for CAMC, respectively, indicating differences in water droplet inclination on these materials. Depending on the investigation, the literature reports that the water’s contact angle on pure chitosan films ranges from 70 to 104° [41]. The contrast in droplet shapes and contact angles offers insight into the hydrophobicity, wetting, and adhesion characteristics of the film. A contact angle of 89.3° for CAMC film indicates a relatively hydrophobic surface, where water beads up, while the reduced contact angle of 75.9° for MC films implies a more hydrophilic surface.

### 3.5. Scanning Electron Microscopy (SEM) Images of MC and CAMC

The surface morphology and roughness of MC and CAMC films are shown in Figure 8. Figure 8a,b depict the SEM micrographs of MC film at two different spatial positions, while Figure 8c,d show SEM micrographs of CAMC film at two different peripheral positions within the sample.

A 10 nm thick platinum layer was putter-coated on the surface of all samples to reduce charging and enable higher-resolution imaging by providing a conductive layer on top of the sample. As shown in Figure 8a,b, the MC film exhibits some pores with sizes ranging from 60 to 80 nm. However, the CAMC film (Figure 8c,d) shows fewer pores, further validating the success of cross-linking, which makes the structure more compact. This result is also supported by the WVP and WVTR values, which are reduced by 28.7% and 10.2%, respectively (Table 2). 

Overall, the surface of both chitosan-based films (MC and CAMC) is smooth, homogenous, and exhibits a continuous matrix with good structural integrity. The surfaces are flat and compact without any phase separation. Moreover, the surface of the CAMC film appears rougher compared to the MC films, possibly due to residual citric acid molecules. SEM micrographs from other studies revealed similar morphologies in chitosan-based composite films [42,43,44].

### 3.6. Packaging Application for Bananas

The performance of the MC and CAMC films' food packaging was evaluated by using banana fruit. Since, in fresh bananas, continuous metabolic activities and respiration occur, allowing the degradation process to be easily monitored through visual changes in their appearance and quality.

Additionally, the loss of moisture from the bananas leads to surface wrinkling due to the loss of cell firmness. The digital photos showing alterations in the degradation of the three groups of bananas—unpackaged, packaged in polyethylene bags, and packaged in CAMC pouches—over 10 days with regular intervals can be observed in Figure 9. The bananas were kept under observation at room temperature.

Initially, on the first day, all three groups of bananas had fresh yellow outer skin with a smooth texture and no speckles (see Figure 9(a1–a3)). Their moisture content was high initially. After storing for ten days to investigate physical changes, the greatest moisture loss was observed in unpackaged bananas (Figure 9(b3–f3)), resulting in a dry and dull texture. This change in quality was also observed, with the color transforming from brilliant yellow to a blackish brown. The banana packaging in polyethylene bags as control samples (Figure 9(b2–f2)) displayed moderate outcomes with slightly yellow and brown in color and exhibited wrinkles and dryness. Moisture loss from the bananas led to surface wrinkling caused by the loss of firmness in the cells. By the fourth day, the bananas’ initially pure yellow color had started changing to brownish yellow with spots and freckles, indicating deterioration during storage. However, the preservation of moisture by the bananas packed in CAMC pouches (Figure 9(b1–f1)) is in good agreement with calculated WVP values, indicating a greater moisture barrier that prevents the escape of moisture from the packaging. As a result, they maintained their vibrant yellow color after 10 days of storage. These results demonstrated that the incorporated CA into chitosan can delay the deterioration of bananas and enhance their shelf life.

Weight loss is a key factor in determining the shelf life of a banana. Figure 10 shows that bananas packaged with CAMC film experienced a significantly lesser loss in weight in comparison to the unpackaged and polyethylene-packaged bananas, with loss in weight readily increasing over the period of storage time. 

Bananas wrapped with CAMC films showed a minimal weight loss of 6.8 ± 0.9% after ten days of storage. On the other hand, bananas packaged in polyethylene bags experienced a weight loss of 21.5 ± 1.5%, while unwrapped bananas showed a higher weight loss of 30.5 ± 1.8%. It is widely thought that fresh fruits and vegetables lose weight through their peel because of vapor pressure, which contributes to the softening of the flesh, fruit ripening, and senescence through metabolic processes. Additionally, weight loss is also caused by the respiration process, as a single carbon atom is shed from the fruit during each cycle.

### 3.7. Comparison with Existing Chitosan-Based Packaging Solutions

CAMC films offer several unique advantages that make them good options for food packaging applications. One key benefit is their sustainable sourcing, as they utilize chitosan extracted from the biomass of *Tenebrio molitor* larvae, promoting circular economy principles and reducing environmental impact. Chitosan extracted from mealworms is a biodegradable polymer, as demonstrated by various studies, highlighting its potential for eco-friendly applications [45,46,47]. The improved water resistance achieved through cross-linking significantly enhances their applicability for food packaging, addressing a common limitation of many biodegradable films. This enhanced barrier property is particularly valuable for preserving food quality and extending shelf life. Additionally, the use of citric acid for cross-linking and glycerol as a plasticizer allows for tailored mechanical properties, potentially offering a balance between strength and flexibility. However, CAMC films also face potential challenges. Their mechanical strength may not match some synthetic polymers like PLA. The production process could be more complex than some alternatives, potentially affecting large-scale manufacturing. Table 3 gives a comprehensive comparison of our fabricated CAMC film with existing chitosan-based films used for packaging applications.

Several studies have explored chitosan-based films with various additives, each achieving distinct properties suited to different packaging applications. Films enhanced with glycerol are known for their excellent barrier properties and flexibility [48]. Other research has focused on incorporating additives like starch/turmeric to impart antibacterial activity [49], or cellulose acetate phthalate/zinc oxide to extend the shelf life of black grapes [50]. Additionally, cellulose content has been shown to influence film characteristics, affecting both thickness and moisture content [51]. Materials such as tannic acid/Moringa oleifera seed powder have successfully increased the shelf life of strawberries [52], while calcium carbonate/chestnut films offered good optical and permeation properties but lacked flexibility [53]. Olive pomace-based films demonstrated effective antioxidant properties, extending the storage life of nuts [54], and films with ascorbate provided green oxidation resistance [55].

In comparison, our work using mealworm shells as the source of chitosan presents several advantages. The film created from mealworm shells, enhanced with glycerol and citric acid, effectively preserves bananas for up to 10 days, aligning with the effective barrier and flexible properties observed in glycerol-enhanced films [48]. The thickness of the films is an essential factor for mechanical and barrier properties. Our CAMC film exhibits a thickness of 0.12 ± 0.05 mm, which is slightly greater than the glycerol/chitosan films (0.06 ± 0.007 mm) [48], but thinner compared to starch/turmeric films (0.28 ± 0.01 mm) [49]. This balanced thickness contributes to a practical combination of durability and flexibility. Moisture content is critical for film stability: the moisture content of our CAMC film is 14.54 ± 0.45%, which is lower than that of the glycerol/chitosan films (19.24 ± 0.23%) and significantly lower than starch/turmeric/chitosan films (38 ± 1%). This lower moisture content suggests improved stability for food preservation. Although the cellulose/chitosan films (18.89 ± 0.94%) have a similar moisture content, our film’s better hydrophobic properties, as indicated by a contact angle of 89.3°, enhance its performance in moisture resistance compared to other films.

The water solubility of chitosan-based films varies significantly depending on the incorporated materials. The water solubility of our CAMC film is 22.8 ± 2.10%, which compares favorably with films incorporating starch/turmeric (19 ± 1%) and tannic acid/Moringa oleifera seed powder (28.5 ± 5.01%). Water vapor permeability (WVP) is vital for food packaging applications to limit the transmission of water vapor molecules in and out of the packaging. The water vapor permeability (WVP) of CAMC film is 2.11 ± 0.32 g/s·m·Pa, demonstrating competitive moisture barrier capabilities when compared to other chitosan-based films, though not as low as the cellulose film (14.80 ± 10^−12^ g/s·m·Pa). This balance suggests that CAMC film is effective in extending shelf life while maintaining flexibility and barrier properties. An additional advantage of our work is the use of mealworm shells as a chitosan source. This approach not only provides a sustainable alternative to traditional shrimp or crab shells but also aligns with current trends toward more eco-friendly materials. The combination of glycerol and citric acid as additives further enhances the film’s properties, a common strategy to improve chitosan film performance for food packaging applications.

The findings indicate that among two prepared films, CAMC (mealworm chitosan cross-linked with citric acid) was found to have the optimal performance, exhibiting good physical and chemical properties along with excellent packaging capabilities. This study suggests that developing chitosan-based films from mealworm biomass cross-linked with citric acid presents significant implications for the food packaging industry and environmental sustainability. This innovative approach addresses two critical environmental issues: reducing plastic waste and upcycling agricultural byproducts. Currently, around 350 to 380 million tons of plastic are produced annually, with packaging contributing the most to plastic waste, making up 61% of it [5]. Conventional plastic packaging contributes significantly to environmental pollution, as it is often non-biodegradable made from chemical and petrochemical-based plastics and accumulates in landfills and oceans. 

As a non-toxic, eco-friendly alternative to chemical preservatives and petrochemical-based plastics, the development of biobased packaging materials, such as CAMC films, offers a sustainable alternative that can help mitigate the environmental impact of conventional plastic waste. Replacing conventional plastic packaging with bio-based sustainable packaging alternatives could significantly reduce plastic waste sent to landfills and prevent it from affecting other ecosystems, such as marine and terrestrial environments [56]. This significant reduction in plastic waste is possible through the adoption of various biodegradable and compostable materials that can effectively replace traditional plastic packaging. The study also highlights the potential for customized packaging solutions and the integration of additives to chitosan, opening avenues for further innovation in sustainable food packaging. Additionally, the use of mealworm biomass from smart farming facilities demonstrates a model for a circular economy in agriculture, promoting resource efficiency and integrated production systems. Furthermore, this research contributes to the growing body of work on bio-based packaging materials, potentially driving a shift toward more sustainable practices in the food packaging industry. By emphasizing the broader implications of CAMC films, we aim to encourage further research and development in this field, ultimately leading to more sustainable and eco-friendly packaging solutions.

The current study has several limitations that need to be addressed in future research. One significant limitation is the scalability of the chitosan extraction process from mealworm biomass. While we have demonstrated the feasibility of extracting chitosan from mealworm shells on a laboratory scale, scaling up this process to industrial levels poses several challenges. These challenges include optimizing the extraction efficiency, reducing costs, and ensuring consistent quality of the extracted chitosan. Future research should investigate methods for large-scale production and assess the economic viability of this approach for commercial applications. According to a report by Grand View Research, the global chitosan market size was valued at USD 10.88 billion in 2022 and is expected to grow at a compound annual growth rate (CAGR) of 20.1% from 2023 to 2030, highlighting the potential market demand for scalable production methods [57]. Additionally, the evaluation period for our study was limited to 10 days. This timeframe, while sufficient to demonstrate the short-term efficacy of CAMC films in preserving bananas, does not provide insights into the long-term stability and performance of the films. Extended studies are necessary to assess how these films behave over longer periods and under various storage conditions, such as fluctuating temperatures and humidity levels. For example, researchers in [45] showed that the long-term durability of biodegradable films can vary significantly based on environmental factors, necessitating comprehensive long-term testing.

Expanding research to encompass a broader range of food products with varying moisture contents, pH levels, and storage requirements would further demonstrate the versatility of these films. For instance, different types of produce such as leafy greens, berries, and citrus fruits may have different preservation needs that our CAMC films could potentially address. Moreover, incorporating various additives into chitosan can further enhance the film’s properties and its suitability for food packaging applications. For example, adding natural antimicrobial agents like essential oils or plant extracts could provide additional protective benefits. A review study by Noori et al. (2023) reported that chitosan-based films incorporated with essential oils exhibited enhanced packaging performance in food models, suggesting a promising area for future research [58]. Developing and evaluating different compositions of chitosan-based films would help determine their effectiveness in preserving various types of food products. Additionally, investigating the impact of different cross-linking agents and plasticizers on the mechanical and barrier properties of the films could lead to further optimization. Lastly, conducting a life cycle assessment (LCA) comparing CAMC films to conventional packaging materials would provide a comprehensive picture of their overall environmental impact. By addressing these limitations and exploring these areas for future research, we can further advance the development of sustainable packaging solutions that are both effective and environmentally friendly.

## 4. Conclusions and Future Recommendations

In this study, we successfully showed how chitosan was extracted from the biomass of yellow mealworm (*Tenebrio molitor*) shell waste by a chemical extraction process comprising three steps: demineralization, deproteinization, and deacetylation. Subsequently, we successfully fabricated mealworm chitosan (MC) and mealworm chitosan cross-linked with citric acid and incorporated with glycerol (CAMC) films by employing the solvent casting technique. FTIR examinations confirmed the successful glycerol incorporation and citric acid cross-linking. We observed a substantial enhancement in the properties of CAMC films while comparing them to those of MC films. The successful cross-linking caused the functionalized films to exhibit enhanced barrier properties, as evidenced by the decrease in water absorption and moisture content values. Additionally, the CAMC films exhibited a remarkable water vapor barrier. The solubility also experienced a decrease, possibly due to the absence of hydrophilic sites for water molecules to bind. Furthermore, the CAMC films exhibited increased flexibility, transparency, and biodegradability, which are essential attributes for food packaging material. When compared to unpackaged bananas or those enclosed in a polyethylene bag, the bananas packaged in CAMC pouches demonstrated extended preservation. The study’s promising results suggest that these films could be effectively utilized as functional packaging pouches to increase the storage time of various fresh produce, not limited to just bananas.

Future study should focus on increasing the production process of CAMC films to determine their commercial viability. Increasing the range of fresh food produce for which these films are used could confirm their efficacy as active functional packaging materials. Additionally, to comprehend the long-term sustainability of using chitosan-based films in comparison to traditional plastic packaging, thorough environmental impact assessments—such as life cycle analyses—are required. Examining the possibility of adding natural additives—like plant extracts and essential oils—could improve the films’ mechanical, antibacterial, and antioxidant qualities and result in stronger, more environmentally friendly packaging options.

## Figures and Tables

**Figure 1 materials-17-03670-f001:**
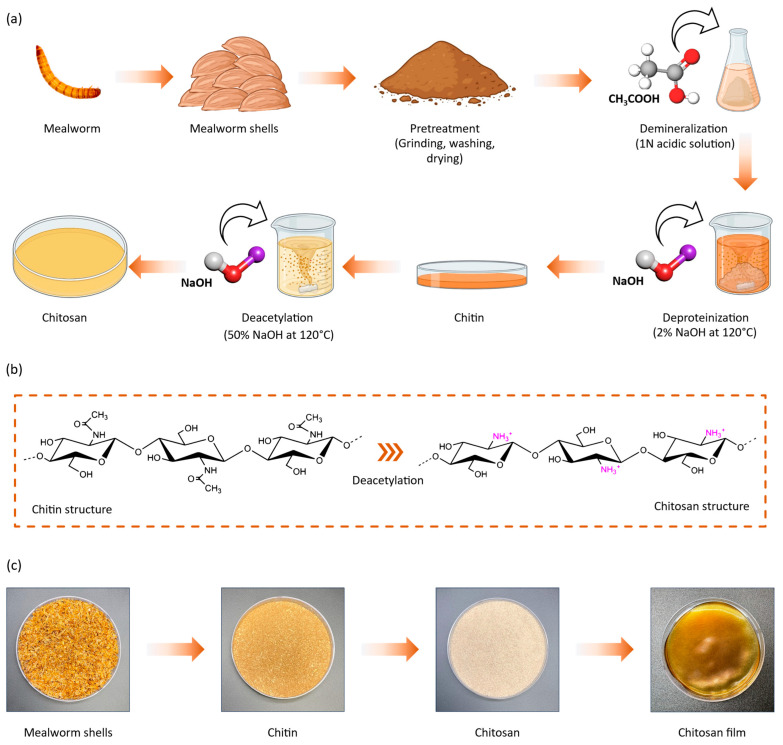
(**a**) Flowchart depicting the sequential steps by which chitin and chitosan are extracted from biomass of yellow mealworm shells; (**b**) Chemical structures of chitin and chitosan; chitin is deacetylated to form chitosan when treated with aqueous solution of sodium hydroxide; (**c**) Digital photos of mealworms, mealworm shells, chitin during the extraction process, and the extracted chitosan, and chitosan film. The film was prepared using the solvent casting method.

**Figure 2 materials-17-03670-f002:**
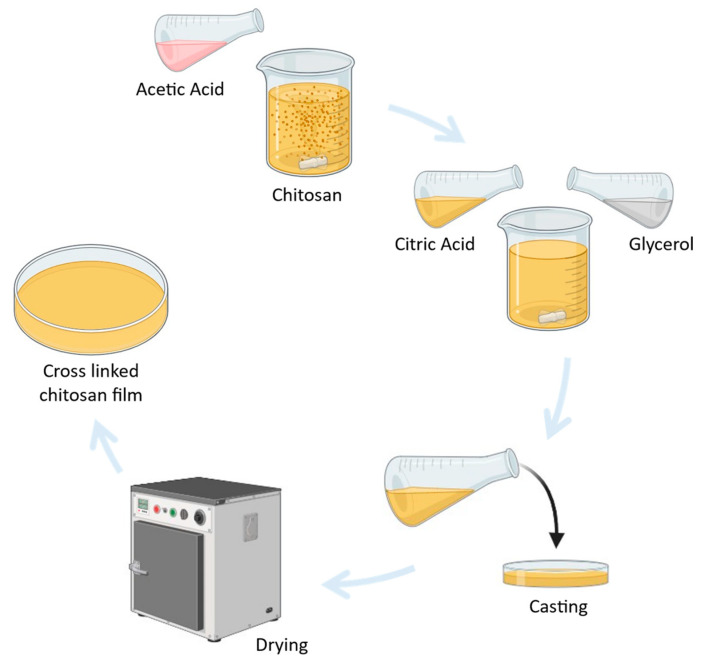
A diagrammatic representation of the casting process used to create a bio composite film.

**Figure 3 materials-17-03670-f003:**
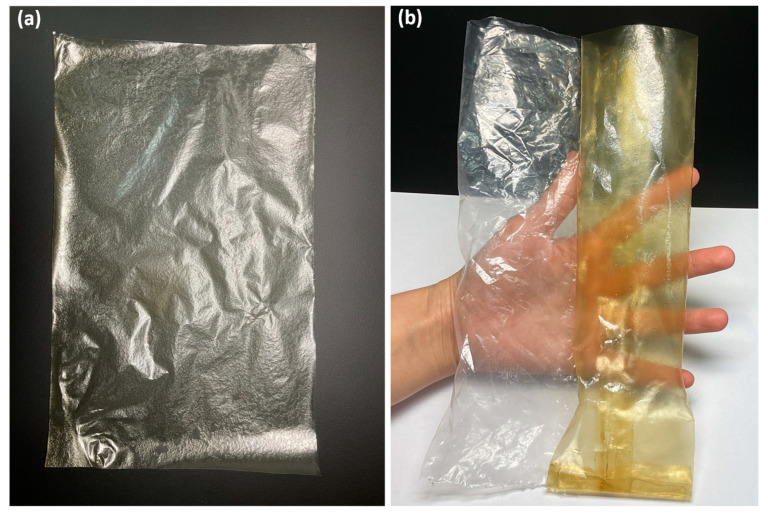
Flexible banana packaging pouches made from (**a**) CAMC film before making pouch; (**b**) prepared polyethylene and CAMC pouches.

**Figure 4 materials-17-03670-f004:**
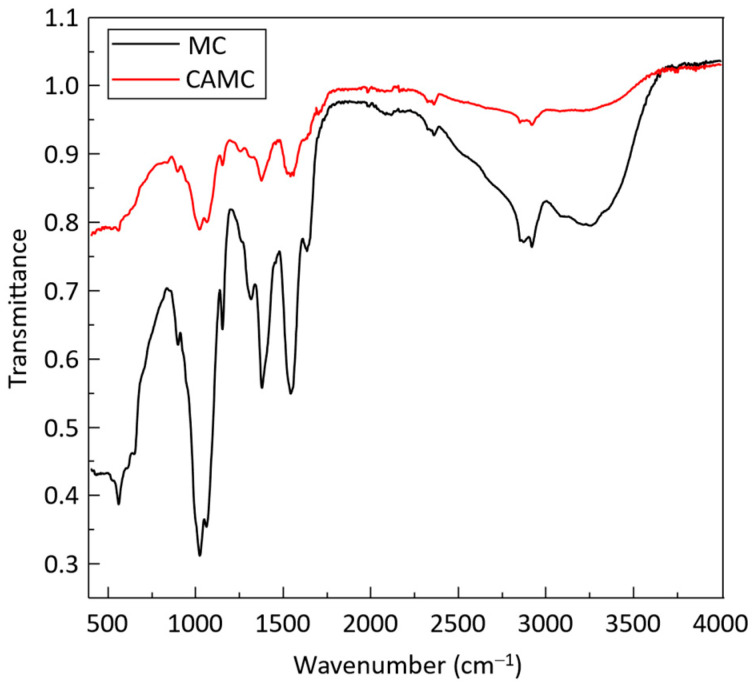
FT-IR spectra of MC and CAMC films.

**Figure 5 materials-17-03670-f005:**
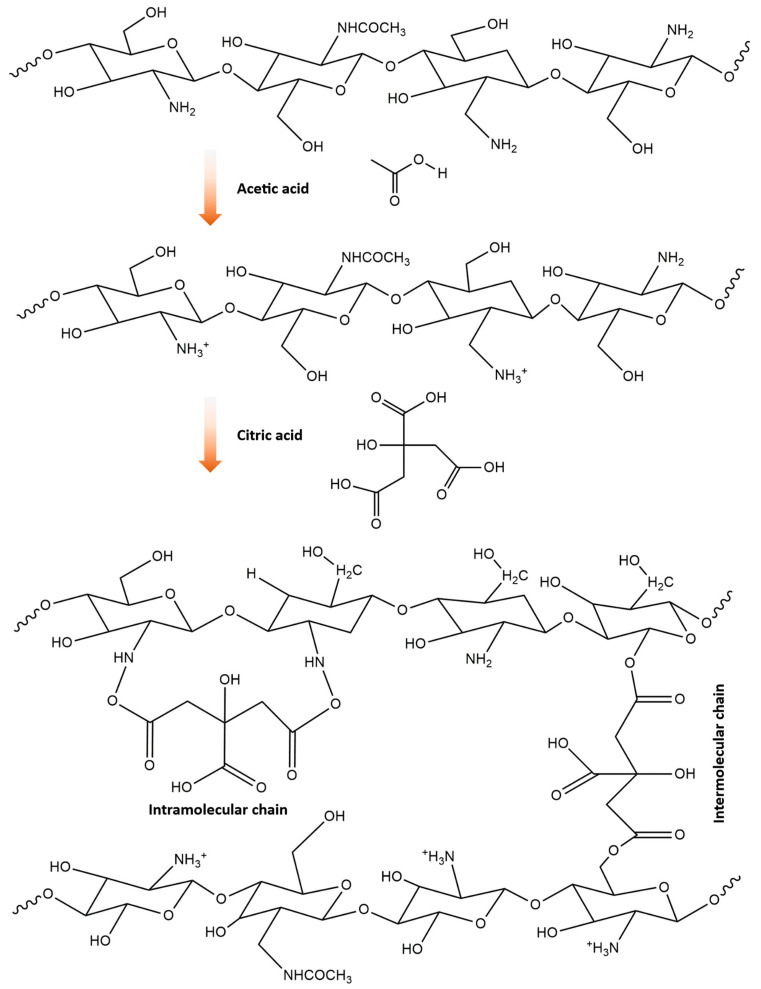
Assumed cross-linking scheme between citric acid and chitosan.

**Figure 6 materials-17-03670-f006:**
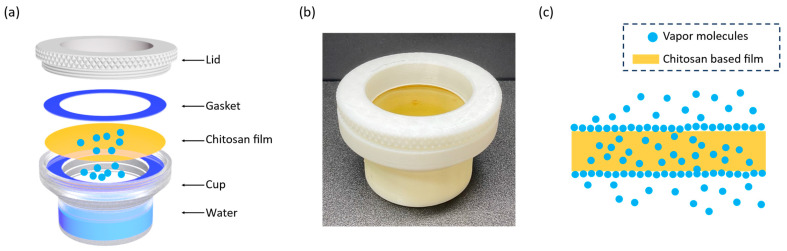
Moisture barrier characteristics of MC and CAMC films: (**a**) An outline of standard ASTM E96 water vapor permeability test using the cup method; (**b**) Digital photo of the chitosan-based film incorporated on the printed water cup for measuring water vapor permeability; (**c**) Illustration showing water vapor molecules passing through the chitosan-based film.

**Figure 7 materials-17-03670-f007:**
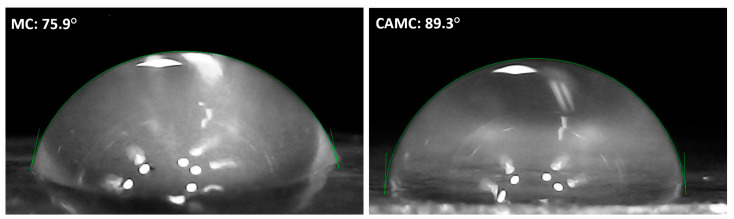
Contact angle measurement of the fabricated films: MC and CAMC.

**Figure 8 materials-17-03670-f008:**
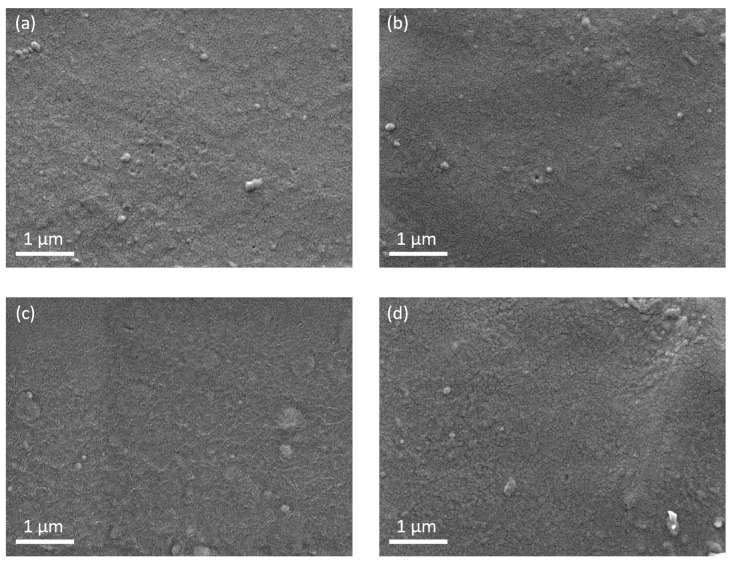
SEM images: (**a**,**b**) of mealworm-extracted chitosan (MC) film; (**c**,**d**) of citric acid cross-linked mealworm chitosan (CAMC) film.

**Figure 9 materials-17-03670-f009:**
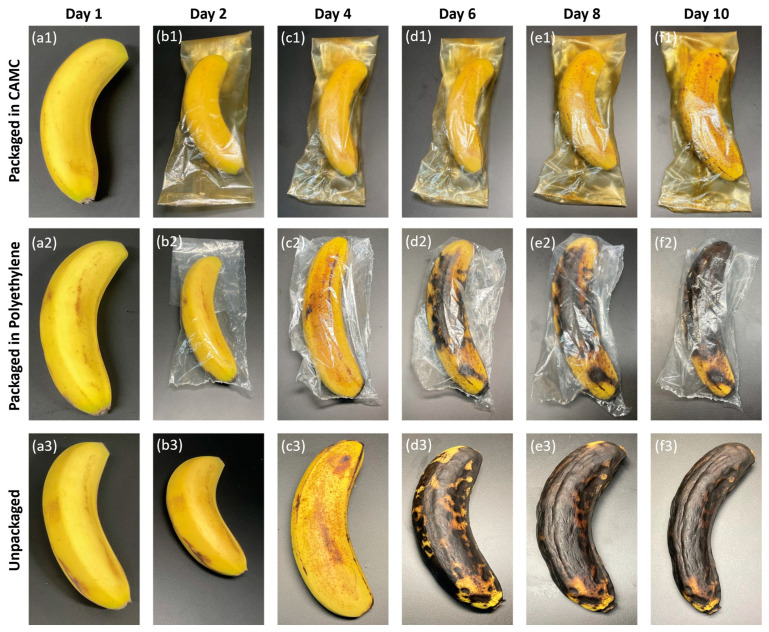
Illustration of degradation of bananas when stored without packaging (**b3**–**f3**), packaged in polyethylene bags (**b2**–**f2**), and packaged in CAMC films (**b1**–**f1**) over 10 days of storage. (**a1**–**a3**) display the bananas on day 1 before starting the experiment.

**Figure 10 materials-17-03670-f010:**
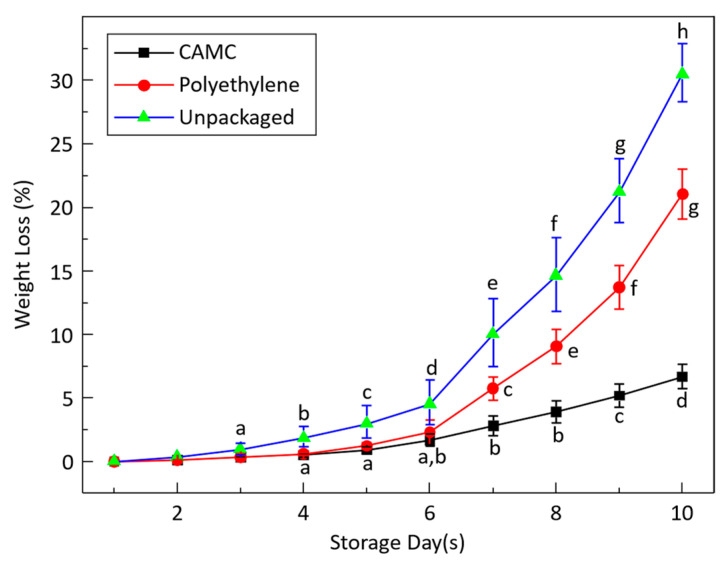
Bananas packaged in polyethylene, CAMC, and unpackaged showed weight loss reduction over a duration of 10 days. Each value represents the average mean of three replicates, and vertical error bars indicate the standard deviation. The letters “a–h” indicated significant differences (*p* < 0.05).

**Table 1 materials-17-03670-t001:** Physicochemical Properties of MC and CAMC film.

Film Kind	Water Solubility (%)	Water Absorption (%)	Moisture Content (%)
MC	37.55 ± 1.54	59.74 ± 3.26	19.99 ± 0.45
CAMC	22.8 ± 2.10	50.1 ± 12.54	14.54 ± 0.78

**Table 2 materials-17-03670-t002:** Moisture barrier parameters of MC and CAMC film.

Type of Film	WVP	WVTR
MC	2.96 ± 0.74	841.8 ± 74
CAMC	2.11 ± 0.32	757 ± 94

**Table 3 materials-17-03670-t003:** Comparison of physical properties of chitosan-based composite films for packaging.

Additive Used with Chitosan	Source of Chitosan	Water Solubility (%)	Film Thickness (mm)	Moisture Content (%)	Water Contact Angle	WVP (g/S·m·Pa)	Performance Highlights	Ref.
glycerol	Shrimp shells	--	0.06 ± 0.007	19.24 ± 0.23	90°	8.07 ± 1.00	Effective barrier and flexible properties	[48]
starch/turmeric	Crab shells	19 ± 1	0.28 ± 0.01	38 ± 1	--	0.31 ± 0.01	Antibacterial activity of chitosan	[49]
cellulose acetate phthalate/zinc oxide	Crab shells	--	0.18± 0.008	--	71°, 81°, 90°	--	Black grape: Increased shelf life up to 9 days	[50]
cellulose	--	28.98 ± 1.45	0.031	18.89 ± 0.94	102°	14.80 × 10^−12^	Viscosity of cellulose affects the food packaging characteristic of chitosan	[51]
Tannic acid/Moringa oleifera seed powder	Shrimp shells	28.5 ± 5.01	0.03	--	77.3°	--	Increased shelf life of Strawberries up to 12 days	[52]
calcium carbonate/chestnut	Shrimp shells	--	0.072	6.0	75°	5.787 × 10^−8^	Good optical and permeation properties but lacks flexibility	[53]
Olive pomace	--	44.55 ± 1.28	0.305 ± 0.00	16.47 ± 0.87	--	2.23 ± 0.14 (g·mm/m^2^·h·kPa)	Showed protective effect against oxidation of nuts for 31 days storage	[54]
Ascorbate	--	36.3 ± 1.0	0.068 ± 8	11.6 ± 0.9	--	6.6 ± 0.2	Green oxidation-resistant packaging material	[55]
glycerol/citric acid (CAMC)	Mealworm shells	22.8 ± 2.10	0.12 ± 0.05	14.54 ± 0.45	89.3°	2.11 ± 0.32	Preserved bananas for 10 days with excellent barrier and flexible properties	Our work

## Data Availability

The original contributions presented in the study are included in the article, further inquiries can be directed to the corresponding authors.

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
