# Peer review of "Chitosan Extracted from the Biomass of Tenebrio molitor Larvae as a Sustainable Packaging Film"

_materials, 2024, doi:10.3390/ma17153670_

Round 1
Reviewer 1 Report
Comments and Suggestions for Authors
In this manuscript the authors developed CAMC films by chitosan extracted from yellow mealworm shells waste, crosslinked with citric acid and incorporated with glycerol. The components were characterized by FTIR. CAMC exhibited improved barrier property and water vapor barrier with decreased water absorption and moisture content values. CAMC films showed enhanced flexibility and transparency for food packaging material and maintained banana for longer time. However, there are some concerns that need to be addressed.
The authors mentioned several times that the CAMC is biodegradable, however, there is lack of experiment result to support this conclusion, although the CAMC is mainly made by chitosan. Please provide more information and evidence to prove that the CAMC is biodegradable.
There is lack of surface characterization of CAMC film such as pore size of the film. I would suggest the authors characterize the surface of CAMC film by SEM.
No error bar and statistical significance shown in Figure 9. Please provide statistical significance and error bar in the figure. If the banana packaging experiment was only conducted with singlet sample, I would suggest the author repeat this experiment with triplet sample size at least.
Please provide the statistical analysis method in the materials and methods section.
Author Response
Respected reviewer, thank you for your valuable feedback. Please find the detailed responses to your comments in the attached document. The corresponding revisions and corrections are highlighted in red with track changes in the revised manuscript.

Reviewer 2 Report
Comments and Suggestions for Authors
This manuscript studied Chitosan extracted from the biomass of Tenebrio molitor larvae as a biodegradable packaging film. It is an interesting paper with practical applications. However, I have several comments as follows;
1. shell concentration of 3%, Is 3% by weight or volume? If it is by weight, should be wt%.
2. with 3% chitin concentration, 3% is by weight?
3. Typo: Line: 467 which were found to be 75.9° for MC and 89.3° for CAMC respectively.
Author Response

(The authors gave the same response as above.)

Reviewer 3 Report
Comments and Suggestions for Authors
See the attachment

Author Response

(The authors gave the same response as above.)

Round 2
Reviewer 1 Report
Comments and Suggestions for Authors
The revised version addressed all of my concerns.